# Dual-Optimized Adaptive Graph Reconstruction for Multi-View Graph Clustering

### Zichen Wen*
School of Computer Science and Engineering, University of Electronic Science and Technology of China
Chengdu, China
zichen.wen@outlook.com

### Tianyi Wu*
School of Computer Science and Engineering, University of Electronic Science and Technology of China
Chengdu, China
tianyi-wu@outlook.com

### Yazhou Ren†
School of Computer Science and Engineering, University of Electronic Science and Technology of China
Chengdu, China
yazhou.ren@uestc.edu.cn

### Yawen Ling
School of Computer Science and Engineering, University of Electronic Science and Technology of China
Chengdu, China
yawen.Ling@outlook.com

### Chenhang Cui
School of Computer Science and Engineering, University of Electronic Science and Technology of China
Chengdu, China
chenhangcui@gmail.com

### Xiaorong Pu
School of Computer Science and Engineering, University of Electronic Science and Technology of China
Chengdu, China
puxiaor@uestc.edu.cn

### Lifang He
Department of Computer Science and Engineering, Lehigh University
Bethlehem, PA, USA
lih319@lehigh.edu

## Abstract

Multi-view clustering is an important machine learning task for multi-media data, encompassing various domains such as images, videos, and texts. Moreover, with the growing abundance of graph data, the significance of multi-view graph clustering (MVGC) has become evident. Most existing methods focus on graph neural networks (GNNs) to extract information from both graph structure and feature data to learn distinguishable node representations. However, traditional GNNs are designed with the assumption of homophilous graphs, making them unsuitable for widely prevalent heterophilous graphs. Several techniques have been introduced to enhance GNNs for heterophilous graphs. While these methods partially mitigate the heterophilous graph issue, they often neglect the advantages of traditional GNNs, such as their simplicity, interpretability, and efficiency. In this paper, we propose a novel multi-view graph clustering method based on dual-optimized adaptive graph reconstruction, named DOAGC. It mainly aims to reconstruct the graph structure adapted to traditional GNNs to deal with heterophilous graph issues while maintaining the advantages of traditional GNNs. Specifically, we first develop an adaptive graph reconstruction mechanism that accounts for node correlation and original structural information. To further optimize the reconstruction graph, we design a dual optimization strategy and demonstrate the feasibility of our optimization strategy through mutual information theory. Numerous experiments demonstrate that DOAGC effectively mitigates the heterophilous graph problem.

## CCS Concepts

• **Mathematics of computing** → **Cluster analysis**; • **Theory of computation** → **Unsupervised learning and clustering**.

## Keywords

Multi-View Graph Clustering, Homophily, Heterophily, Graph Reconstruction, Graph Neural Networks.

**ACM Reference Format:**
Zichen Wen, Tianyi Wu, Yazhou Ren, Yawen Ling, Chenhang Cui, Xiaorong Pu, and Lifang He. 2024. Dual-Optimized Adaptive Graph Reconstruction for Multi-View Graph Clustering. In *Proceedings of the 32nd ACM International Conference on Multimedia (MM '24), October 28-November 1, 2024, Melbourne, VIC, Australia.* ACM, New York, NY, USA, 12 pages. https://doi.org/10.1145/3664647.3680677

*Equal Contribution.
†Corresponding author.

## 1 Introduction

Clustering is a fundamental unsupervised learning task with broad applications across various fields [14, 18, 56]. Multi-view clustering (MVC) has obtained considerable interest owing to its capacity to harness information from multiple views [40, 50], thereby enhancing clustering performance [41, 42, 59, 63, 67]. Over the past few years, many MVC methods have been proposed, which can be generally classified into three primary categories [60]: co-training

approaches [6, 12, 24, 70, 72], low-rank matrix factorization techniques [48, 65, 66, 68], and subspace-based methods [7, 15, 19, 37, 45, 46, 51, 52, 61]. Nevertheless, they often fail to effectively utilize the common multi-view graph-structured data. With the development of graph neural networks (GNNs) [11, 13, 44], researchers have been interested in utilizing GNNs to extract the abundant structural information embedded in graph data [47]. However, labeling graph data becomes increasingly challenging as the amount of graph data grows. Therefore, multi-view graph clustering (MVGC) has emerged as a popular and valuable research area. Many GNN-based approaches have been proposed and have effectively advanced the development of MVGC. For example, Fan et al. [9] develop a one2multi graph autoencoder clustering framework (O2MAC) to capture the shared feature representation. Hassani and Ahmadi [16] propose to learn node and graph level representations by contrasting structural views of graphs. However, traditional GNNs are typically designed for homophilous graphs, where edges connect nodes of the same class. As a result, existing GNN-based MVGC methods are less effective when applied to heterophilous graphs, where edges connect nodes of diverse classes. In reality, heterophilous graph data is prevalent. For instance, in the context of protein chemistry, interactions often occur between different types of amino acids [2, 73]. In financial transaction networks, fraudulent users frequently engage in transactions with non-fraudulent users [39]. Additionally, dating networks often exhibit a higher number of connections between individuals of opposite genders [39, 69].

To address this challenge, several techniques have been introduced to improve GNNs for heterophilous graphs. These novel GNN variants aim to overcome the limitations of traditional GNNs, which rely on neighborhood aggregation mechanisms. They can be roughly divided into two groups [69]: non-local neighbor extension methods [1, 29, 36, 73] and GNNs architecture refinement methods [5, 29, 62]. Most of these methods enable newly designed GNNs to partially address the issue of heterophilous graphs by aggregating feature information from higher-order neighbors [17] or adapting the internal GNN structure. However, these methods increase the computational complexity of the model and even may degrade the performance of homophilous graph data [39] after structural modification to accommodate heterophilous graph data. Meanwhile, Li et al. [26] also point out that traditional GNNs have advantages in terms of simplicity [55], explainability [55], and efficiency [64] that GNN variants cannot match.

In addition, some studies have attempted to explore the reasons for the poor performance of traditional GNNs in dealing with heterophilous graphs and propose novel solutions from the point of the spectral domain [22]. For example, Bo et al. [2] design a mechanism that can integrate low-frequency signals, high-frequency signals, and raw features. Liu et al. [31] propose a novel graph representation learning method with edge heterophily discriminating (GREET) that learns representations by discriminating and leveraging homophilous edges and heterophilous edges. Luan et al. [32] propose adaptive channel mixing to exploit local and node-wise information from three channels: aggregation, diversification, and identity. Wen et al. [53] propose an adaptive hybrid graph filter related to homophily degree that adaptively captures low and high-frequency information. These spectral domain filtering methods aim to capture rich information in every frequency band of the graph to acquire distinguishable node representations. Inevitably, however, to capture information in various frequency bands, these methods usually design multiple filters or design filters with multiple channels. Undoubtedly, training multiple filters will increase the training cost as it multiplies the parameters. Several studies have pointed out that traditional GNNs based on the homophily assumption are actually low-pass filters spectrally [25, 34]. In other words, designing diverse filters to capture graph signals in multiple frequency bands is actually equivalent to modifying GNNs in the spatial domain, which would also suffer from the same drawbacks as the previously mentioned GNN variants. Considering that traditional GNNs mining graph structure information still has advantages in some aspects and there are drawbacks in transforming GNNs in spatial and spectral domains, we propose to reconstruct the original graph structure so that the reconstructed graphs can be adapted to the traditional GNNs, as a way to solve the problem of heterophilous graphs in MVGC.

Our motivation is to reconstruct the graph that can be applied to message passing and neighborhood aggregation mechanisms of GNNs. To achieve this goal, we propose a dual-optimized adaptive graph reconstruction method. To be specific, we first construct the node correlation matrix. Although directly utilizing the node correlation matrix as a reconstruction graph can improve the homophily degree as shown in Table 3 of Appendix B, the node correlation matrix is only constructed based on the node feature information, which completely discards the original structural information of the graph, resulting in suboptimal performance. Taking into account both the degree of homophily and original structural information, we propose an adaptive mechanism for reconstructing the graph. Specifically, we utilize pseudo-labeling information to quantify the homophily degree of the original adjacency matrix. Based on this, the weight of the original adjacency matrix is assigned in the reconstruction graph to selectively preserve a certain amount of original structural information when the graph type is unknown.

To further optimize the graph structure, we develop a dual optimization strategy for the autoencoder. The first optimization comes from the autoencoder's reconstruction loss function, which can compress and denoise the data while preserving valid information about the input data. Furthermore, a random mask is applied to the original node feature information leading to the creation of an additively noisy feature matrix. Next, GNNs' message passing and neighborhood aggregation mechanisms are utilized to recover the noisy feature, followed by the use of a noise recovery loss function to minimize any differences between the recovered feature information and the original feature information. However, trainable parameters are not set for the GNN applied to recover feature information. Instead, only its aggregation mechanism is utilized. The training objective has now shifted to the autoencoder, and the noise recovery loss function directly propagates the gradient back to the autoencoder, optimizing its training process.

In summary, our main contributions are as follows:

- To alleviate the poor performance of GCN on heterophilous graphs, we design an adaptive graph reconstruction mechanism, employing the pseudo-labeling information.
- We devise a dual optimization strategy for reconstruction graphs, which makes reconstruction graphs more adaptable to neighborhood aggregation mechanisms.

- We demonstrate the feasibility of using the processing of noisy node feature recovery to assist the GCN aggregation process based on mutual information.
- Experimental results on real-world datasets and synthetic datasets indicate that our approach achieves state-of-the-art performance on most evaluation metrics.

## 2 Related Works

### 2.1 Multi-View Graph Clustering

Recently, researchers have been interested in utilizing GNNs to extract structural information from graphs. Numerous multi-view graph clustering methods have been proposed. Fan et al. [9] design a one2multi graph autoencoder to capture shared feature representation. Cheng et al. [4] propose two-pathway graph encoders to map graph embedding features and learn view-consistency information. Xia et al. systematically explore the cluster structure using a graph convolutional encoder trained to learn the self-expression coefficient matrix [57]. In addition to designing diverse graph encoders, contrastive learning methods are employed to extract information from graphs. Hassani and Ahmadi [16] introduce a self-supervised model to learn the node representations by contrasting structural views of graphs. Pan and Kang [35] employ contrastive learning to uncover the shared geometry and semantics in order to learn a consensus graph. Additionally, Lin and Kang utilize graph filtering techniques to smooth the features and learn a consensus graph for clustering [27]. Zhou and Du [71] enhance clustering by learning a consensus graph filter from multiple data views. Despite the attractive performance of these methods, they are often sensitive to the quality of graph structure. In other words, they generally do not perform well with heterophilous graphs.

### 2.2 Heterophilous Graph Representation Learning

Several efforts have been extended to address the issue of heterophilous graphs. Chien et al. [5] tackle heterophilous graph issues by employing GNNs that propagate using specially learnable weights. Chanpuriya and Musco [3] develop a feature extraction technique capable of adapting to graph structures exhibiting both homophily and heterophily. Li et al. propose an innovative graph restructuring approach that extends spectral clustering through alignment with node labels [26]. However, applying them to MVGC poses challenges as they heavily depend on true node label information. Additionally, there are unsupervised techniques aimed at addressing the heterophilous graph issue that do not depend on true labeling information, such as GREET [31], which discriminates between homophilous and heterophilous edges using an edge discriminator, enabling separate processing of these edges. Xiao et al. propose a decoupled self-supervised learning framework to decouple various underlying semantics among different neighborhoods [58]. While these methods partially address the heterophilous graph challenge, they are difficult to generalize to MVGC since these methods are designed for node classification tasks. The lack of a feasible and effective solution to mitigate the negative impact of heterophilous information still persists for heterophilous graphs in multi-view graph clustering.

## 3 Methodology

### 3.1 Preliminaries

In the task of multi-view graph clustering, the objective is to group a set of $n$ nodes into $k$ clusters. To achieve this, we utilize the notation $\mathcal{G} = (\mathcal{V}, \mathcal{E})$ to denote a graph. Here, $\mathcal{V}$ represents the nodes set, and the set of all nodes belonging to class $i$ is represented as $\mathcal{V}_i$, with $N = |\mathcal{V}|$, and $\mathcal{E} \subseteq \mathcal{V} \times \mathcal{V}$ represents the edge set with self-loops. The feature matrix for the nodes is denoted as $\mathbf{X} \in \mathbb{R}^{N \times d}$, and the symmetric adjacency matrix of the graph $\mathcal{G}$ is represented by $\mathbf{A} \in \mathbb{R}^{N \times N}$, with elements $a_{ij} = 1$ indicating the presence of an edge between node $i$ and node $j$, and $a_{ij} = 0$ otherwise. Additionally, we define the degree matrix of $\mathbf{A}$ as $\mathbf{D}_{ii}^v = \sum_j a_{ij}^v$, enabling the normalization of each view's $\mathbf{A}^v$ to $\widetilde{\mathbf{A}}^v = (\mathbf{D}^v)^{-1}\mathbf{A}^v$. The normalized graph Laplacian matrix, denoted as $\widetilde{\mathbf{L}}^v$, is then calculated as $\mathbf{I} - \widetilde{\mathbf{A}}^v$, with $\mathbf{I}$ representing the identity matrix.

### 3.2 Adaptive Graph Construction

To ensure the adaptability of GCN's neighborhood aggregation mechanism, it is necessary for a majority of the neighboring nodes in the reconstruction graph to be of the same class as the central node. Nodes belonging to the same class tend to have similar node feature vectors. Therefore, we prioritize the feature information of the nodes and aim to construct the graph by mining the correlation between their features. In this way, the resulting graph aligns with our expectation of connecting nodes to neighboring nodes that share the same label.

Firstly, we harness the remarkable capabilities of autoencoder to extract node feature information and refine the original features of the nodes within the graph:

$$\mathbf{Z}^v = f^v(\sigma(\mathbf{X}; \mathbf{W}_\theta)), \quad (1)$$

$$\bar{\mathbf{X}}^v = g^v(\sigma(\mathbf{Z}^v; \mathbf{W}_\varphi)), \quad (2)$$

where $\mathbf{Z}^v \in \mathbb{R}^{N \times d_{Z^v}}$, $v \in V$. $\mathbf{W}_\theta$ and $\mathbf{W}_\varphi$ represent the learnable parameters of the encoder and decoder in the $v$-th view respectively, and $\sigma(\cdot)$ is the activation function.

After this, we explore the correlation among nodes by computing the cosine similarity between nodes features and derive the correlation matrix $\mathbf{S}^v$:

$$\mathbf{S}^v = Sim(\mathbf{Z}^v, \mathbf{Z}^{vT}) = \frac{\mathbf{Z}^v \cdot \mathbf{Z}^{vT}}{\|\mathbf{Z}^v\| \cdot \|\mathbf{Z}^{vT}\|}, \quad (3)$$

where $Sim(\cdot)$ represents the cosine similarity function in vector space. Intuitively, if node $i$ and node $j$ belong to the same class, then $\mathbf{S}_{ij}^v$ will have a larger value in the correlation matrix $\mathbf{S}^v$, indicating that nodes $i$ and $j$ are similar and connected.

Directly using $\mathbf{S}^v$ as a reconstruction graph improves the homophily degree of the graph to some extent. Nevertheless, only utilizing the node feature information and totally disregarding the original structural information of the graph may not be entirely beneficial to our graph reconstruction. Therefore, we propose a selective utilization of the original graph structure information to reconstruct graphs while maintaining a high degree of homophily. To accomplish this, we design an adaptive reconstruction graph mechanism. Specifically, we attempt to quantify the degree of homophily in the original graph structure and subsequently assign appropriate weights. However, in the unsupervised context, access

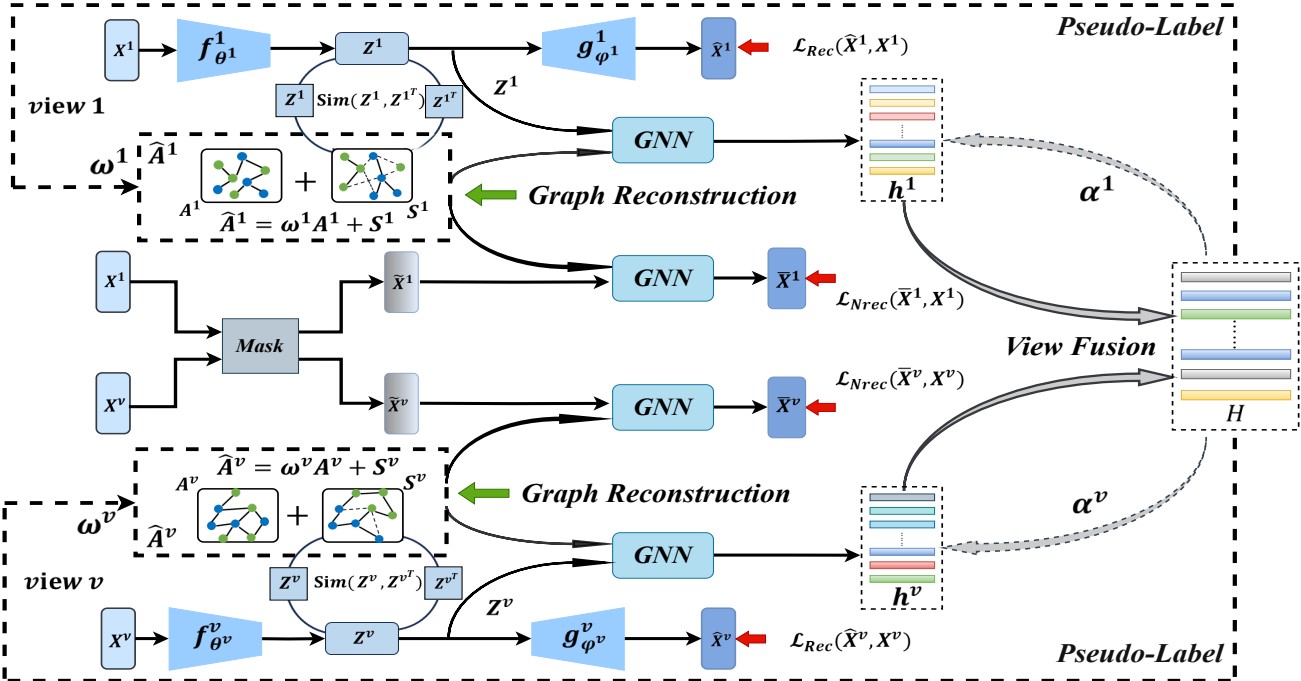

**Figure 1: The framework of our DOAGC model. The inputs to each view are the node feature matrix X and the original adjacency matrix A. The output is the consensus embedding H fused by each view node embedding h, after which H is used as $k$-means input for clustering.**

to true labeling information is not available. For this reason, we choose to approximate the homophily degree of the original graph structure by using the pseudo-labeling information obtained from the final consensus embedding **H** of Eq. (12):

$$w^v = \frac{\sum_{i,j}(\mathbf{A}_{i,j}^v \odot \bar{Y}_i \bar{Y}_j^T - \mathbf{I}_{i,j})}{\sum_{i,j}(\mathbf{A}_{i,j}^v - \mathbf{I}_{i,j})}, \qquad (4)$$

where $w^v$ denotes the original graph weight obtained after homophily degree computation from the last iteration, $\odot$ represents the Hadamard product, $\bar{Y} \in \{0, 1\}^{n \times c}$ is the pseudo label obtained from the clustering of consensus embedding **H**.

Finally, we obtain the reconstruction graph that incorporates the assessment of the homophily level and original structural data:

$$\hat{\mathbf{A}}^v = \mathbf{S}^v + w^v \mathbf{A}^v. \qquad (5)$$

As shown in Fig. 2, $w^v$ can converge from different initialization values to the same value that is close to the true homophily degree through the adaptive mechanism, which indicates that the adaptive mechanism is stable and meets our expectations.

## 3.3 Dual Optimization Strategy for Reconstruction Graph

There is a significant disparity between solely reconstructing graphs based on extracting correlation information among nodes and our objective, which is to construct graphs optimized for GCN neighborhood aggregation mechanism. Therefore, we devise a dual optimization strategy to improve the reconstruction graph in Section 3.2.

Specifically, we first design the reconstruction loss function of the autoencoder utilizing the cross-entropy loss:

$$\mathcal{L}_{Rec} = \sum_{v=1}^{V} l(\hat{\mathbf{X}}^v, \mathbf{X}^v) = -\sum_{v=1}^{V} \sum_{i,j} (\hat{x}_{ij}^v \cdot \log(x_{ij}^v)). \qquad (6)$$

As the first optimization of the reconstruction graph, the reconstruction loss of the autoencoder mainly ensures the validity of the information extracted from the node feature matrix, i.e., **Z** is able to reflect the essential attributes of the nodes, and the valid information of the node features will not be lost in the process of dimensionality reduction and denoising.

Furthermore, we design the second optimization process. Specifically, we begin by adding a random mask as noise to the original matrix **X** of node features. Then, we utilize GCN's neighborhood aggregation mechanism to recover the feature information following the addition of noise, which can be represented from $\tilde{\mathbf{X}} \xrightarrow{AGG_{\hat{A}}} \bar{\mathbf{X}} \rightarrow \mathbf{X}$ (**Process 2**), where $\tilde{\mathbf{X}}$ denotes the nodes features with random mask (noisy features), $\bar{\mathbf{X}}$ means the nodes features recovered by the aggregation mechanism of GCN and $AGG_{\hat{A}}$ denotes the aggregation operation of GCN using reconstruction graph. The noise recovery loss is defined as follows:

$$\mathcal{L}_{Nrec} = \sum_{v=1}^{V} l(\bar{\mathbf{X}}^v, \mathbf{X}^v) = -\sum_{v=1}^{V} \sum_{i,j} (\bar{x}_{ij}^v \cdot \log(x_{ij}^v)), \qquad (7)$$

where $\bar{\mathbf{X}}^v = GCN(\hat{\mathbf{A}}^v, \tilde{\mathbf{X}}^v)$.

In fact, the second optimization process attempts to optimize the reconstruction graph in the form of supervised learning. Due to the unknowability of labels in unsupervised tasks, we cannot directly

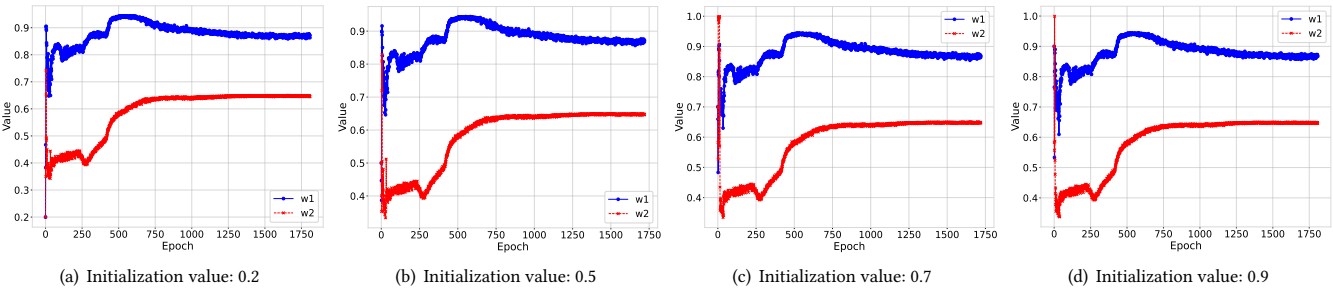

(a) Initialization value: 0.2    (b) Initialization value: 0.5    (c) Initialization value: 0.7    (d) Initialization value: 0.9

Figure 2: The adaptive process of $w$ on ACM.

utilize the true label information to supervise the reconstruction of graph, making the aggregated features $\mathbf{h}$ approximate the true label Y, i.e., $\mathbf{X} \xrightarrow{AGG_{\hat{A}}} \mathbf{h} \rightarrow \mathbf{Y}$ (**Process 1**).

Intuitively, we believe that a graph suitable for predicting node features is also suitable for predicting node labels. Therefore, we introduce **Process 2** to enhance **Process 1**. As shown below, we utilize mutual information theory to speculate on the validity and rationality of **Process 2**. Specific symbol explanations are shown in Table 1 of Appendix A.

LEMMA 3.1. [10] *Let* $\tilde{\mathbf{X}}_{\bar{\mathbf{y}}_i}$ *and* $\tilde{\mathbf{X}}_{\mathbf{y}_i}$ *be mutually redundant for* $x_i$*, i.e., the feature of node i , where* $\tilde{\mathbf{X}}_{\bar{\mathbf{y}}_i}$ *and* $\tilde{\mathbf{X}}_{\mathbf{y}_i}$ *denotes the nodes features with random masks belonging to different and the same class as node* $i$ *respectively. The recovered feature* $\bar{x}_i$ *is aggregated from* $\tilde{\mathbf{X}}_{\mathbf{y}_i}$*. If* $\bar{x}_i$ *is sufficient for* $\tilde{\mathbf{X}}_{\bar{\mathbf{y}}_i}$ *(*$I(\tilde{\mathbf{X}}_{\bar{\mathbf{y}}_i}; \tilde{\mathbf{X}}_{\mathbf{y}_i} | \bar{x}_i) = 0$*), the mutual information between* $x_i$ *and* $\bar{x}_i$ *have the following relationship:*

$$I_\theta(x_i; \bar{x}_i) = I_\theta(x_i; \tilde{\mathbf{X}}_{\mathbf{y}_i}, \tilde{\mathbf{X}}_{\bar{\mathbf{y}}_i}) = I_\theta(x_i; \tilde{\mathbf{X}}), \qquad (8)$$

*where* $\theta$ *denotes the learnable parameters of the autoencoder. The proof is given in Appendix A.*

The left side of Eq. (8) is the object we need to optimize, i.e., $\mathcal{L}_{Nrec}$ in Eq. (7).

Similarly, replacing the symbol of **Process 2** in Eq. (8) with the corresponding symbol of **Process 1**, we have:

$$I_\theta(y_i; h_i) = I_\theta(y_i; \mathbf{X}), \qquad (9)$$

**Hypothesis 1**: The distribution of node features X can be seen as a joint distribution of fragmented information of node label y ($\tilde{Y}$) and random noise ($e$).

Based on **Hypothesis 1**, we have:

$$\begin{aligned}
I_\theta(y_i; h_i) &= I_\theta(y_i; \mathbf{X}) \\
&= I_\theta(y_i; \tilde{\mathbf{Y}}, e) \\
&= I_\theta(y_i; \tilde{\mathbf{Y}}) + I_\theta(y; e | \tilde{\mathbf{Y}}) \\
&= I_\theta(y_i; \tilde{\mathbf{Y}}).
\end{aligned} \qquad (10)$$

Eq. (10) implies that the process of aggregating node features in GCN, with the goal of maximizing the mutual information between $y_i$ and $h_i$ ($I_\theta(y_i; h_i)$), can be viewed as a way of consolidating fragmented label information.

Then, we obtain two formulas: $I_\theta(x_i; \bar{x}_i) = I_\theta(x_i; \tilde{\mathbf{X}})$ and $I_\theta(y_i; h_i) = I_\theta(y_i; \tilde{\mathbf{Y}})$ that correspond to **Process 2** and **Process 1**, respectively. We can also observe similarity between the two processes: they are both the process of integrating fragmented information

that belongs to the same category. These two processes share same graph structure, and the elements involved can all correspond one-to-one. So, if we can train **Process 2** well and optimize the reconstruction graph i.e., minimizing $\mathcal{L}_{Nrec}$, the Eq. (10), alternatively, the aggregation process of GCN, can also benefit from it.

## 3.4 View Weighting and Fusion

In a multi-view task, different views contain not exactly the same information, i.e., there is consistency and complementarity among the views [21, 54]. To fully utilize the complementary information among views, we attempt to get a consensus embedding containing rich information by fusing the node embedding $\mathbf{h}^v$ of each view [20]. However, to account for the varying information values of different views, it is essential to assign suitable weights to each view based on their quality evaluation. This weighting scheme ensures that different views make distinct contributions to the final consensus embedding. We first obtain the node embedding for each view:

$$\mathbf{h}^v = GCN(\hat{\mathbf{A}}^v, \mathbf{Z}^v). \qquad (11)$$

Naturally, it occurs to us to utilize the obtained consensus embedding H to in turn guide the embedding $\mathbf{h}^v$ of each view to assign weights to it. Specifically, if a view's embedding $\mathbf{h}^v$ is similar to the consensus embedding, then the information it carries must be important and we assign larger weight to it, and vice versa. We obtain the consensus embedding H as follows:

$$\mathbf{H} = \sum_{v=1}^{V} \alpha^v \mathbf{h}^v, \qquad (12)$$

where $\alpha^v$ denotes the weight of the node embedding for the $v$-th view and is calculated as follows:

$$\alpha^v = \left( \frac{eva^v}{\max(eva^1, eva^2, \cdots, eva^V)} \right)^\rho. \qquad (13)$$

Here $eva^v$ is obtained from the evaluation function that computes the similarity between the consensus embedding H and each view embedding $\mathbf{h}^v$, i.e., $eva^v = evaluation(\mathbf{h}^v, \mathbf{H})$. The hyperparameter $\rho$ is used to adjust the degree of smoothing or sharpening of the view weights. For the final consensus embedding H, we apply the $k$-means algorithm to get the clustering results.

## 4 Experiments

## 4.1 Evaluation Setup and Metrics

*4.1.1 Datasets.* To evaluate the effectiveness of the proposed method, we conducted experiments on nine graph datasets with

**Table 1: The detailed statistics information of the six graph datasets.**

| Datasets | ACM | DBLP | Minesweeper | Cornell | Chameleon | Wisconsin |
|---|---|---|---|---|---|---|
| Nodes | 3,025 | 4,057 | 10000 | 183 | 2,277 | 251 |
| Features | 1,830 | 334 | 7 | 1,703 | 2,325 | 1,703 |
| Clusters | 3 | 4 | 2 | 5 | 5 | 5 |
| Graphs | $\mathcal{G}_1, \mathcal{G}_2$ | $\mathcal{G}_1, \mathcal{G}_2, \mathcal{G}_3$ | $\mathcal{G}_1, \mathcal{G}_2$ | $\mathcal{G}_1, \mathcal{G}_2$ | $\mathcal{G}_1, \mathcal{G}_2$ | $\mathcal{G}_1, \mathcal{G}_2$ |
| Homophily degree | 0.82, 0.64 | 0.80, 0.67, 0.32 | 0.68, 0.68 | 0.30, 0.30 | 0.23, 0.23 | 0.19, 0.19 |

**Table 2: The clustering results on six real-world datasets. The best results are shown in bold, and the second-best results are underlined. All experimental results were averaged after performing the experiment five times and the hyperparameter settings for all baseline models followed the recommendations in their respective original papers.**

| Method/Datasets | ACM | | | | DBLP | | | | Minsweeper | | | |
|---|---|---|---|---|---|---|---|---|---|---|---|---|
| | NMI% | ARI% | ACC% | F1% | NMI% | ARI% | ACC% | F1% | NMI% | ARI% | ACC% | F1% |
| VGAE (2016) | 49.1 | 54.4 | 82.2 | 82.3 | 69.3 | 74.1 | 88.6 | 87.4 | 4.1 | **8.9** | 69.7 | 60.1 |
| DAEGC (2019) | 63.8 | 70.1 | 89.0 | 88.9 | 30.8 | 33.4 | 66.5 | 65.6 | 5.1 | 3.1 | 58.9 | 55.2 |
| AGE (2020) | 73.5 | 78.9 | 92.4 | 92.4 | 45.0 | 47.6 | 75.3 | 74.6 | **6.2** | 4.6 | 60.7 | 60.7 |
| O2MAC (2020) | 69.2 | 73.9 | 90.4 | 90.5 | 72.9 | 77.8 | 90.7 | 90.1 | 2.9 | 1.6 | 58.3 | 53.9 |
| MvAGC (2020) | 67.4 | 72.1 | 89.8 | 89.9 | 77.2 | 82.8 | 92.8 | 92.3 | 0.5 | −1.3 | 58.8 | 46.5 |
| AGCN (2021) | 68.4 | 74.2 | 90.6 | 90.6 | 39.7 | 42.5 | 73.3 | 72.8 | 0.0 | −2.1 | 60.6 | 46.7 |
| MCGC (2021) | 71.3 | 76.3 | 91.5 | 91.6 | **83.0** | 77.5 | 93.0 | 92.5 | 0.3 | −1.7 | 66.3 | 47.1 |
| DCRN (2022a) | 71.6 | 77.6 | 91.9 | 91.9 | 49.0 | 53.6 | 79.7 | 79.3 | 1.2 | 4.5 | 64.4 | 54.6 |
| DuaLGR (2023) | 73.2 | 79.4 | 92.7 | 92.7 | 75.5 | 81.7 | 92.4 | 91.8 | 0.2 | −0.3 | 60.0 | 47.8 |
| DOAGC (ours) | **78.2** | **83.5** | **94.2** | **94.3** | 79.5 | **84.3** | 93.4 | 92.9 | 0.4 | −1.6 | **78.5** | **78.5** |

| Method/Datasets | Cornell | | | | Chameleon | | | | Wisconsin | | | |
|---|---|---|---|---|---|---|---|---|---|---|---|---|
| VGAE (2016) | 7.6 | 11.2 | 53.4 | 26.8 | 15.1 | 12.4 | 35.4 | 29.6 | 10.5 | 13.7 | 49.3 | 34.1 |
| DAEGC (2019) | 7.4 | 3.8 | 35.0 | 28.2 | 9.1 | 5.6 | 32.2 | 31.2 | 10.6 | 3.4 | 32.7 | 28.3 |
| AGE (2020) | 9.6 | 7.8 | 43.2 | 43.2 | 8.6 | 7.6 | 32.4 | 32.4 | 9.3 | 1.3 | 31.1 | 31.1 |
| O2MAC (2020) | 5.6 | 4.1 | 42.3 | 26.4 | 12.3 | 8.9 | 33.5 | 28.6 | 11.0 | 8.9 | 40.0 | 27.9 |
| MvAGC (2020) | 10.0 | 0.1 | 45.5 | 19.2 | 10.8 | 3.3 | 29.2 | 24.3 | 8.1 | 4.8 | 47.7 | 20.6 |
| AGCN (2021) | 5.0 | 2.5 | 56.3 | 19.9 | 6.7 | 6.1 | 32.5 | 20.4 | 6.4 | 6.8 | 49.8 | 24.9 |
| MCGC (2021) | 7.7 | 9.2 | 55.7 | 29.6 | 9.5 | 5.9 | 30.0 | 19.1 | 12.9 | 5.9 | 51.8 | 30.7 |
| DCRN (2022a) | 20.5 | 32.8 | 66.1 | 40.5 | 8.7 | 5.7 | 30.9 | 21.9 | 10.8 | 16.0 | 50.2 | 34.1 |
| DuaLGR (2023) | 28.5 | 22.4 | 57.0 | 41.0 | 19.5 | 16.0 | 41.1 | 37.7 | 34.1 | 28.8 | 56.4 | 47.1 |
| DOAGC (ours) | **43.1** | **46.4** | **73.2** | **45.1** | **22.1** | **18.5** | **44.2** | **40.4** | **55.5** | **57.4** | **79.7** | **54.5** |

different homophily degrees. **ACM** [9] is derived from the ACM database[1] and is composed of two graphs: the co-paper network and the co-subject network. **DBLP** [9], sourced from the DBLP database[2], consists of three graphs: co-author, co-conference, and co-term. **Minesweeper** is a synthetic graph emulating the eponymous game [39]. **Wisconsin** and **Cornell** [36] are webpage graphs from WebKB[3] and **Chameleon** is a subset of the Wikipedia network [43]. The detailed statistics of the datasets are presented in Table 1 and Appendix B.

*4.1.2 Evaluation Metrics.* We utilize accuracy (ACC), normalized mutual information (NMI), adjusted rand index (ARI), and F1-score (F1) to evaluate the clustering performance of the proposed model.

*4.1.3 Comparison Methods.* To validate the superiority of the proposed method, we utilize popular benchmarks for comparative experiments and analysis. VGAE [23] and AGE [8] represent

two distinct graph encoding techniques. DAEGC [49] is a goal-directed deep attentional embedded graph clustering framework. O2MAC [9] is an approach that acquires information from both node features and graph structures. MvAGC [4] and MCGC [35] represent two recent graph-based methods that utilize graph filtering to acquire a consensus graph. AGCN [38] is an attention-driven graph clustering network. DCRN [30] is a method that improves the performance of graph clustering by reducing the information correlation. DualGR [28] utilizes soft-labels and pseudo-labels to provide guidance in the process of refining and fusing graphs for clustering.

## 4.2 Performance Comparison

Table 2 presents the clustering performance of all compared methods on six real-world graph datasets. From the results, we can see that DOAGC demonstrates competitive performance. Specifically, when facing graph datasets with a high homophily degree, such as ACM (HR **0.82** & **0.64**), DBLP (HR **0.80** & **0.67** & **0.32**), and Minesweeper (HR **0.68** & **0.68**), DOAGC on ACM outperforms the SOTAs in ACC, NMI, ARI, and F1 by 1.5%, 4.7%, 4.1%, and 1.6%,

---
[1]https://dl.acm.org/
[2]https://dblp.uni-trier.de/
[3]http://www.cs.cmu.edu/afs/cs.cmu.edu/project/theo-11/www/wwkb

**Table 3: The clustering results on six synthetic ACM graph datasets with different homophily degrees. The best results are shown in bold. And the second-best results are underlined.**

| Method/Datasets | ACM00 (HR 0.00 & 0.00) | | | | ACM01 (HR 0.10 & 0.10) | | | | ACM02 (HR 0.20 & 0.20) | | | |
|---|---|---|---|---|---|---|---|---|---|---|---|---|
| | NMI% | ARI% | ACC% | F1% | NMI% | ARI% | ACC% | F1% | NMI% | ARI% | ACC% | F1% |
| VGAE (2016) | 0.5 | 0.5 | 37.4 | 37.1 | 0.5 | 0.5 | 37.1 | 35.6 | 0.4 | 0.4 | 36.9 | 34.9 |
| DAEGC (2019) | 43.5 | 46.4 | 77.5 | 76.1 | 19.8 | 22.5 | 64.0 | 63.5 | 5.0 | 5.5 | 43.6 | 43.2 |
| AGE (2020) | 0.0 | 0.0 | 33.5 | 33.5 | 0.0 | 0.0 | 34.3 | 33.9 | 0.1 | 0.0 | 34.9 | 34.7 |
| O2MAC (2020) | 25.0 | 24.7 | 55.0 | 54.6 | 17.6 | 17.1 | 49.9 | 49.7 | 9.6 | 9.4 | 42.9 | 42.8 |
| MvAGC (2020) | 0.9 | 0.9 | 37.1 | 35.5 | 1.9 | 2.0 | 40.9 | 39.1 | 5.3 | 5.6 | 45.7 | 45.4 |
| AGCN (2021) | 0.8 | 0.8 | 38.7 | 38.5 | 0.7 | 0.7 | 36.4 | 36.2 | 4.1 | 4.4 | 44.7 | 44.5 |
| MCGC (2021) | 49.8 | 42.9 | 63.0 | 53.5 | 52.9 | 44.7 | 63.9 | 54.6 | 29.1 | 31.7 | 67.7 | 67.2 |
| DuaLGR (2023) | 55.1 | 60.7 | 84.8 | 84.5 | 55.9 | 61.7 | 85.3 | 85.0 | 59.2 | 66.0 | 87.3 | 87.1 |
| DOAGC (ours) | **63.0** | **70.4** | **89.2** | **89.2** | **63.4** | **70.7** | **89.3** | **89.3** | **63.3** | **70.7** | **89.3** | **89.3** |
| Method/Datasets | ACM03 (HR 0.30 & 0.30) | | | | ACM04 (HR 0.40 & 0.40) | | | | ACM05 (HR 0.50 & 0.50) | | | |
| VGAE (2016) | 0.7 | 0.7 | 38.0 | 37.6 | 9.7 | 8.1 | 48.4 | 49.0 | 26.2 | 27.0 | 65.9 | 66.4 |
| DAEGC (2019) | 3.8 | 4.1 | 45.4 | 45.2 | 19.3 | 22.6 | 64.8 | 64.9 | 41.4 | 48.4 | 79.7 | 79.6 |
| AGE (2020) | 0.2 | 0.1 | 35.1 | 35.0 | 13.5 | 15.5 | 50.9 | 48.4 | 24.1 | 21.5 | 59.2 | 57.1 |
| O2MAC (2020) | 6.7 | 6.5 | 40.7 | 40.5 | 5.5 | 5.4 | 40.3 | 40.2 | 6.6 | 6.7 | 42.7 | 42.6 |
| MvAGC (2020) | 15.4 | 16.5 | 57.7 | 57.7 | 36.9 | 39.5 | 74.0 | 74.2 | 64.6 | 71.1 | 89.4 | 89.4 |
| AGCN (2021) | 1.2 | 1.1 | 38.9 | 39.0 | 0.1 | 0.0 | 34.8 | 34.8 | 1.5 | 1.5 | 40.3 | 40.4 |
| MCGC (2021) | 51.8 | 57.2 | 83.0 | 82.9 | 83.9 | 88.8 | 96.2 | 96.2 | 91.0 | 94.4 | 98.1 | 98.1 |
| DuaLGR (2023) | **60.2** | **67.6** | **88.0** | **88.0** | **85.1** | **90.1** | **96.6** | **96.6** | 97.8 | 98.9 | 99.6 | 99.6 |
| DOAGC (ours) | 57.4 | 64.8 | 86.9 | 86.9 | 72.8 | 79.1 | 92.5 | 92.4 | **98.2** | **99.1** | **99.7** | **99.7** |

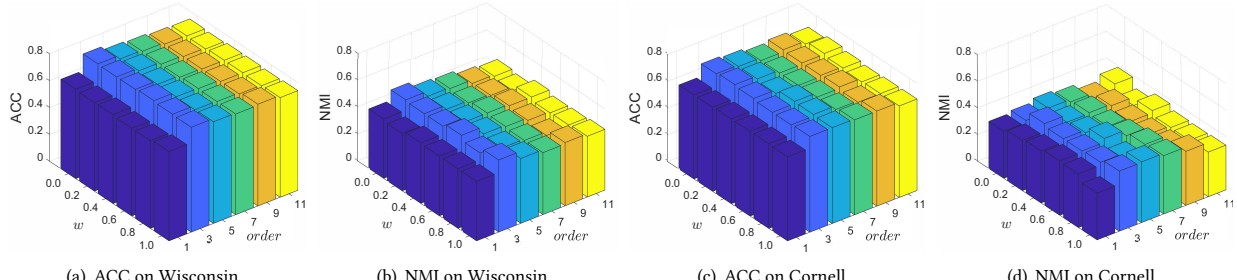

| (a) ACC on Wisconsin. | (b) NMI on Wisconsin. | (c) ACC on Cornell. | (d) NMI on Cornell. |
|---|---|---|---|

**Figure 3: Sensitive analysis of ACC and NMI on Wisconsin and Cornell with *order* and *w*.**

respectively. Meanwhile, it surpasses others on most metrics in both DBLP and Minesweeper. Specifically, it increases ACC, ARI, and F1 on DBLP by 1.0%, 2.6%, and 1.1%, respectively, and ACC, F1 on Minesweeper by 8.8%, and 17.8%, respectively. Furthermore, unlike the poor performance of other baselines on heterophilous graph datasets, DOAGC achieves excellent performance on graphs with low homophily degree. The ACC of our model reaches 79.7% on Wisconsin, while the second-best DualGR [28] is only 56.4%, which appears similar on Cornell and Chameleon.

Table 3 demonstrates the comparison results on six synthetic ACM datasets, and the results show that DOAGC also performs well on the same dataset with different homophily degrees. Comparing DOAGC with other baselines on diverse homophily degrees, our method effectively addresses the challenge faced by previous graph clustering approaches on heterophilous graphs. This enables traditional GNNs, relying on homophily assumptions, to fully leverage structural information mining on heterophilous graphs.

## 4.3 Ablation Study

*4.3.1 Effect of Each Loss.* To explore the importance and effectiveness of each loss function for the proposed model, we removed each loss function separately to observe the change in clustering performance. The detailed data on the ablation experiments for the loss function is presented in Table 4. As indicated in Table 4, both the reconstruction loss $\mathcal{L}_{Rec}$ and the noise recovery loss $\mathcal{L}_{Nrec}$ affect the model's performance. The reconstruction loss $\mathcal{L}_{Rec}$ plays a dominant role in the model's performance, and removing it would lead to a significant decrease in performance. Due to the fact that $\mathcal{L}_{Rec}$ and $\mathcal{L}_{Nrec}$ optimize the autoencoder for training by returning the training gradient to it, in other words, both losses have an optimizing effect, but the $\mathcal{L}_{Rec}$ has greater optimization intensity.

*4.3.2 Effect of Each Component.* To investigate how the reconstruction graph $\hat{\mathbf{A}}^v$ affects the model's performance, we conducted an in-depth ablation analysis of the reconstruction graphs. Specifically, we remove $\mathbf{S}^v$ and $\mathbf{A}^v$ in the reconstruction graph respectively. As shown in Table 3, the model's performance is impacted by the removal of either $\mathbf{S}^v$ or $\mathbf{A}^v$. Removing $\mathbf{S}^v$ has a greater impact on the

**Table 4: The ablation study results of DOAGC on Wisconsin and Cornell. The original results are shown in bold.**

| Compenents / Datasets | Wisconsin | | | | Cornell | | | |
|---|---|---|---|---|---|---|---|---|
| | NMI% | ARI% | ACC% | F1% | NMI% | ARI% | ACC% | F1% |
| DOAGC (w/o $\mathcal{L}_{Rec}$) | 45.5 | 47.2 | 65.3 | 50.2 | 31.6 | 29.7 | 61.7 | 45.1 |
| DOAGC (w/o $\mathcal{L}_{Nrec}$) | 50.4 | 53.2 | 77.7 | 53.3 | 39.1 | 45.2 | 71.6 | 43.4 |
| DOAGC (w/o $\mathbf{S}^v$) | 10.4 | 10.6 | 49.4 | 35.1 | 10.0 | 14.9 | 52.1 | 31.3 |
| DOAGC (w/o $\mathbf{A}^v$) | 51.8 | 52.9 | 77.6 | 49.8 | 38.3 | 45.1 | 71.6 | 40.4 |
| **DOAGC (ours)** | **55.5** | **57.4** | **79.7** | **54.5** | **43.1** | **46.4** | **73.2** | **45.1** |

model's performance compared to removing $\mathbf{A}^v$. $\mathbf{S}^v$ is constructed using node feature information based on the cosine similarity between each pair of nodes. Intuitively, due to the high probability that the nodes with similar feature information belong to the same class, the constructed $\mathbf{S}^v$ has a greater edge weight among similar nodes, i.e., a higher homophily degree. Removing $\mathbf{S}^v$ will greatly reduce the homophily degree of the reconstruction graph, leading to the inability of the GCN message passing mechanism to function, which in turn leads to a decrease in the model performance. $\mathbf{A}^v$ is the original adjacency matrix, and although the removal of $\mathbf{A}^v$ still allows the reconstruction graph to maintain a high homophily degree, the complete abandonment of the original structural information still has an impact on the model performance.

*4.3.3* ***Convergence Analysis.*** We performed experiments on eight real-world datasets with varying degrees of homophily. Fig. 4 shows the detailed results of the experiments about convergence analysis, where the left subgraph refers to the ACM, DBLP, and IMDB with a high homophily degree; the right subgraph refers to the EAT, Texas, Chameleon, Wisconsin, and Cornell with a low homophily degree. As depicted in Fig. 4, the three real-world datasets ACM, DBLP, and IMDB, start to converge before the 50th epoch. During the initial stage of iteration, the loss values significantly decrease. In contrast, for the heterophilous graph datasets such as Texas and Chameleon, they reach the convergence state at epoch = 100, and the loss values gradually decrease thereafter. Overall, our proposed model achieves faster convergence and a significantly reduced loss value in both homophilous and heterophilous graphs, validating the reliability of our proposed model and the efficacy of our dual-optimization training approach.

*4.3.4* ***Parameter Sensitivity Analysis.*** Figure 3 illustrates the parameter sensitivity analysis for *order* and *w*. *order* denotes the degree of GCN neighborhood aggregation, where a higher *order*

implies that the node can aggregate information from more distant nodes. *w* represents the initial weight of adjacency matrix $\mathbf{A}^v$ in Eq. (5). A higher value of *w* indicates that the reconstructed graph $\hat{\mathbf{A}}^v$ contains more of the original structural information at the beginning of the iteration. As shown in subfigures (a) and (c) of Fig. 3, we perform experiments on the Wisconsin and Cornell datasets using different combinations of *order* and *w* to examine the variations in ACC. As observed, fixed *order* renders initial *w* insignificant. DOAGC shows consistent and stable performance regardless of weights in the initial adjacency matrix $\mathbf{A}^v$. Regarding *order*, DOAGC achieves excellent performance at *order* = 3. Further increasing *order* does not improve the model's performance and may even result in a slight decline. This indicates that our method does not require high-order aggregation with increased complexity. Moreover, it suggests that the reconstructed graph $\hat{\mathbf{A}}^v$ predominantly consists of neighboring nodes with the same classifications.

For subfigures (b) and (d) in Fig. 3 related to NMI, the NMI on Wisconsin parallels the description of ACC on Wisconsin above. Cornell's NMI demonstrates slight sensitivity to its parameters, yielding higher values at $order \in \{3, 5, 7\}$, and lower values when the *order* is either too low or too high. When *order* is low, nodes struggle to aggregate extensive neighbor information. Conversely, with a high *order*, nodes tend to aggregate feature information from distant nodes. However, in our reconstruction graph $\hat{\mathbf{A}}^v$, distant high-order neighbor nodes are more likely to belong to different classes, resulting in the aggregation of conflicting information. This, in turn, contributes to a decrease in NMI.

## 5 Conclusion

In this paper, we address the challenge of heterophilous graphs in MVGC and propose DOAGC, a dual-optimization adaptive graph reconstruction multi-view clustering method. DOAGC focuses on reconstructing graphs to facilitate the message passing and neighbor aggregation mechanisms of conventional GNNs. We extract node correlation from feature information and introduce an adaptive mechanism utilizing pseudo-labeling information derived from consensus embedding. Additionally, we propose a dual optimization strategy to enhance the compatibility of the reconstructed graph with traditional GNNs. The efficacy of the optimization strategy is validated through mutual information theory. Our proposed approach achieves outstanding results on eight real-world datasets and six synthetic datasets with varying homophily degrees, providing evidence that DOAGC effectively addresses the heterophilous graph problem encountered by MVGC, while simultaneously maintaining exceptional clustering performance on homophilous graphs.

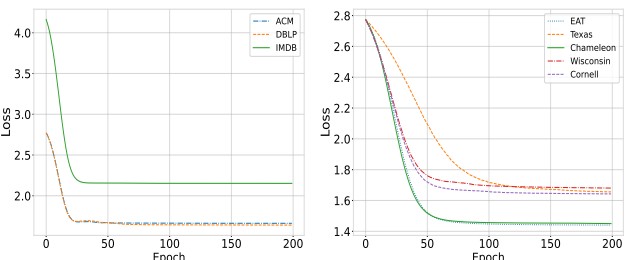

**Figure 4: Convergence analysis.**

# Acknowledgments

This work was supported in part by Sichuan Science and Technology Program (No. 2024NSFSC1473) and in part by Shenzhen Science and Technology Program (No. JCYJ20230807115959041). Lifang He is partially supported by the NSF grants (MRI-2215789, IIS-1909879, IIS-2319451), NIH grant under R21EY034179, and Lehigh's grants under Accelerator and CORE.

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
