# OpenReview forum: "Dual-Optimized Adaptive Graph Reconstruction for Multi-View Graph Clustering"
_acmmm.org/ACMMM/2024/Conference — MM2024 Poster_

### Official Review · Reviewer_rrG6 · 2024-05-24

**Rating:** 5
**Confidence:** 4

**Summary:**

The authors introduce a novel DOAGC method tailored for multi-view graph clustering (MVGC), specifically addressing the challenges posed by heterophilous graphs. The primary objective of DOAGC is to reconfigure the graph structure to better align with conventional Graph Neural Networks (GNNs). It features an adaptive graph reconstruction mechanism utilizing a dual optimization strategy.

**Strengths:**

- The methodology is illustrated in detail, with clear explanations of the adaptive graph construction and dual optimization strategy.
- Experimental results demonstrate DOAGC's effectiveness in addressing issues related to heterophilous graphs, outperforming other baseline methods on various widely used datasets.
- The inclusion of theoretical justifications, such as mutual information theory, adds depth to the understanding of why the proposed method is effective.

**Limitations:**

- The authors should explain how to determine the number of views to be utilized.
- In Table 1, the authors mention the "homophily degree" of the analyzed graphs; however, they do not provide a clear explanation of this term.
- Discussing some recent works on multi-view GNNs in Related Work would be helpful.

**Suitability:**

3

---

### Official Review · Reviewer_CSZr · 2024-05-24

**Rating:** 6
**Confidence:** 3

**Summary:**

The paper proposes a novel multi-view graph clustering method named Dual-Optimized Adaptive Graph Clustering (DOAGC). This approach leverages adaptive graph reconstruction and a dual optimization strategy to address the heterophilous graph issue while maintaining the benefits of traditional GNNs.

**Strengths:**

- The introduction of a dual-optimized adaptive graph reconstruction mechanism is innovative and addresses the limitations of existing GNNs on heterophilous graphs effectively.
- The authors provide extensive experimental results on both real-world and synthetic datasets, showcasing the method's robustness and effectiveness across different scenarios.
- The paper is well organized and well written.

**Limitations:**

- It seems that the proof of mutual information for the dual optimization strategy in the appendix is somewhat confusing. I suggest simplifying the notation for clarity.

**Suitability:**

3

---

### Official Review · Reviewer_YrVc · 2024-05-24

**Rating:** 5
**Confidence:** 3

**Summary:**

To alleviate the poor performance of GCN on heterophilous graphs, an adaptive graph reconstruction mechanism is proposed in this paper. A dual optimization strategy for reconstruction graphs is also designed to make reconstruction graphs more adaptable to neighborhood aggregation mechanisms.

**Strengths:**

(1) The application of mutual information theory to validate the dual optimization strategy provides a robust theoretical foundation for the proposed method.

(2) Experimental results show the superiority of the proposed method on a number of real-world and synthetic datasets.

**Limitations:**

(1) The specific details for training the baseline methods/models are not fully explained.

(2) In Section 3.1, the notation of v is not defined, yet it is directly used as a superscript in lines 287-289.

(3) It is better to add illustrations (e.g., toy example) to explain the motivation of the proposed model.

**Suitability:**

3

---

### Official Review · Reviewer_mZWR · 2024-05-28

**Rating:** 5
**Confidence:** 4

**Summary:**

The motivation of this article is to reconstruct graphs that can be effectively applied to the message passing and neighborhood aggregation mechanisms of GNNs, addressing the issue of GNNs being unsuitable for widely prevalent heterophilous graphs. An adaptive graph reconstruction mechanism is proposed to not only consider node feature information but also preserve the structural information of the original graph. A dual optimization strategy is proposed to compress and denoise data through the reconstruction loss of the autoencoder.

**Strengths:**

1. The use of pseudo labels to calculate the homophily degree of the original adjacency matrix is reasonable and interesting.
2. The dual optimization strategy for reconstructing graphs is well-designed.
3. Experiments on both real-world and synthetic datasets demonstrate that the proposed method has significant advantages over baselines. The ablation experiment further proves the effectiveness of each component of this method.

**Limitations:**

1. Fig. 2 requires more detailed explanations. Specifically, it should clarify how it demonstrates that "w^v can converge from different initialization values to the same value, which is close to the true homophily degree."
2. Some typos should be corrected, such as the inconsistent number of real-world datasets mentioned in the Introduction and Conclusion sections.

**Suitability:**

3

---

### Official Review · Reviewer_JLhh · 2024-05-29

**Rating:** 4
**Confidence:** 4

**Summary:**

In this paper, the authors propose a novel multiview graph clustering method based on dual-optimized adaptive graph reconstruction, named DOAGC. In this framework, an adaptive graph reconstruction mechanism that accounts for node correlation and original structural information is developed. To further optimize the reconstruction graph, a dual optimization strategy is devised, and theorectial analysis is provided to demonstrate the feasibility of our optimization strategy.

**Strengths:**

S1. Overall, the paper is easy to read.

S2. The idea of this paper is interesting, and theorectial analysis is provided to demonstrate the feasibility of our optimization strategy.

S3. Extensive experiments on several benchmarks are made to validate the effectiveness of the proposed method.

**Limitations:**

However, there are still some concerns about this paper,
1. The topic of this paper is meaningful. Concerned about this problem, DualGR[21] is recently proposed. What about the differences between DOAGC and DualGR. To some extend, there are some components similar with DualGR. Actually, the problems/motivations mentioned in this paper are not clear enough.

2. The paper is well-written, but the overall flowchart of DOAGC is a little unsmooth. Moreover, the pseudo code is missing.

3. Figure 1 is not clear to direactly show the idea of DOAGC, and more introductions should be made in the caption.

4. How does the process 1 reflect in Figure 1? The existence of process 1 is to demonstrate the rationality of Process 2?

5. What is the used evaluation function in this paper, i.e., 𝑒𝑣𝑎𝑙𝑢𝑎𝑡𝑖𝑜𝑛($h^𝑣$,H).

6. As shown in Table 4, why only choose Wisconsin and Cornell to do such ablation studies? Additionally, the neccessity of L𝑁𝑟𝑒𝑐 seems not so important. The author are suggested to provide more anaysis about  L𝑁𝑟𝑒𝑐 in this paper.

**Suitability:**

2

---

### Meta-Review · Area_Chair_4giN · 2024-07-01

**Recommendation:** Accept (Poster)
**Confidence:** 5

**Metareview:**

This paper studies heterophilous graph issue in multi-view graph clustering while striving to maintain the simplicity, interpretability, and efficiency of traditional GNNs. It introduces an adaptive graph reconstruction mechanism for node correlation, and provides a dual optimization scheme to update the reconstruction graph. After rebuttal and discussion, all reviewers recognize the contributions and give positive scores. So, acceptance is recommendated.